# NLRP3 Negative Regulation Mechanisms in the Resting State and Its Implications for Therapeutic Development

**DOI:** 10.3390/ijms25169018

**Published:** 2024-08-20

**Authors:** YeJi Kim, Sumin Lee, Yong Hwan Park

**Affiliations:** 1Department of Microbiology, Ajou University School of Medicine, Suwon 16499, Republic of Korea; ispy22@ajou.ac.kr (Y.K.); sumin0038@ajou.ac.kr (S.L.); 2Department of Biomedical Sciences, Graduate School of Ajou University, Suwon 16499, Republic of Korea

**Keywords:** NLRP3 inflammasome, indirect PRRs, pyrin inflammasome, negative regulation, cage model

## Abstract

The NACHT-, leucine-rich-repeat-, and pyrin domain-containing protein 3 (NLRP3) is a critical intracellular sensor of the innate immune system that detects various pathogen- and danger-associated molecular patterns, leading to the assembly of the NLRP3 inflammasome and release of interleukin (IL) 1β and IL-18. However, the abnormal activation of the NLRP3 inflammasome has been implicated in the pathogenesis of autoinflammatory diseases such as cryopyrin-associated autoinflammatory syndromes (CAPS) and common diseases such as Alzheimer’s disease and asthma. Recent studies have revealed that pyrin functions as an indirect sensor, similar to the plant guard system, and is regulated by binding to inhibitory 14-3-3 proteins. Upon activation, pyrin transitions to its active form. NLRP3 is predicted to follow a similar regulatory mechanism and maintain its inactive form in the cage model, as it also acts as an indirect sensor. Additionally, newly developed NLRP3 inhibitors have been found to inhibit NLRP3 activity by stabilizing its inactive form. Most studies and reviews on NLRP3 have focused on the activation of the NLRP3 inflammasome. This review highlights the molecular mechanisms that regulate NLRP3 in its resting state, and discusses how targeting this inhibitory mechanism can lead to novel therapeutic strategies for NLRP3-related diseases.

## 1. Introduction

The inflammasome is a key component of the innate immune system, recognizing pathogen-associated molecular patterns (PAMP) and damage-associated molecular patterns (DAMP) to activate caspase-1, which secretes inflammatory cytokines interleukin (IL) 1β and IL-18 and induces pyroptosis. Several inflammasomes, including nucleotide-binding domain, leucine-rich repeat receptor (NLR) and pyrin domain-containing protein (NLRP) 1, NLRP3, NLR family CARD domain containing protein4 (NLRC4), pyrin, and absent in melanoma 2 (AIM2) have been identified. Although all inflammasomes induce IL-1β and IL-18 secretion by activating caspase-1, each responds to various stimuli. Most inflammasomes sense their targets directly through interaction. For example, NLRC4 recognizes flagellin via the NLR family of apoptosis inhibitory proteins (NAIP) and AIM2 directly detects bacterial DNA. However, some inflammasomes such as pyrin and NLRP3 indirectly recognize bacterial infections by responding to post-infection intracellular changes. The guard hypothesis suggests that this indirect recognition occurs after pathogen effector molecules enter and modify host proteins sensed by specific detectors called guards. This indirect recognition system is well developed in plant cells because of its inability to produce various receptors through somatic mutations, thus defending against various pathogens with limited receptors. The guard system in mammalian cells was not well understood until the activation mechanism of the pyrin inflammasome was discovered in 2014 and 2016, which highlighted the regulatory mechanism of indirect sensors in mammalian cells. This review discusses the activation mechanism of the pyrin inflammasome and the regulatory mechanism of NLRP3, focusing on the suppression of NLRP3 activity in the resting state based on recent studies.

## 2. Plant Guard Hypothesis

Plants, unlike mammals, lack an adaptive mobile immune system. Therefore, each plant cell must detect and respond independently to various pathogens. To avoid pathogen infection, plants rely on two major branches of their immune system: pathogen-associated molecular pattern-triggered immunity (PTI) and effector-triggered immunity (ETI) [1]. Transmembrane pattern recognition receptors (PRRs) recognize PAMPs and activate PTI, which serve as the first barrier of the plant immune system in restricting pathogen growth. Upon sensing PAMPs, PRRs activate downstream immune signaling, including an increase in cytosolic Ca^2+^, reactive oxygen species (ROS), plasma membrane depolarization, and transcriptional or physiological changes [2]. However, many pathogens employ strategies to subvert PTI downstream signaling by secreting virulence factors called effector proteins into the cytosol of host cells. Effectors have diverse functions, such as hijacking the cytoskeletal machinery to create a more hospitable environment for the pathogen, or blocking host translation. Plants have evolutionarily conserved receptors called resistance (R) proteins that sense the changes caused by these effectors and activate ETI. The best-studied R family is the NLR class. NLR proteins recognize effectors directly or indirectly by detecting changes in host proteins and activating ETI. The plant guard hypothesis explains how various effectors can be detected by a single R protein, and how relatively few R proteins can recognize a broad diversity of pathogenic invaders [3] (Figure 1, left).

### 2.1. Pyrin Inflammasome

Pyrin, encoded by *MEFV* gene, is a cytosolic PRR that forms an inflammasome complex in response to bacterial infections. Mutations in the *MEFV* gene cause autoinflammatory diseases, such as Familial Mediterranean Fever (FMF). In 2014, Shao et al. reported that the pyrin inflammasome senses bacterial modifications of Rho GTPases [4]. Pathogenic bacteria survive inside host cells by modifying and inactivating Rho GTPases, which are key regulators of the cytoskeleton. In the resting state, RhoA GTPase activates the serine/threonine kinase PKN1/2, which phosphorylates pyrin at serine 208 and 242 [5]. Phosphorylated pyrin binds to 14-3-3 proteins and inhibits pyrin inflammasome assembly. When bacteria invade mammalian cells, they secrete toxins that inactivate the RhoA GTPases. Upon the inactivation of RhoA, these toxins reduce pyrin phosphorylation, leading to decreased 14-3-3 binding and increased pyrin inflammasome assembly. Thus, pyrin does not directly bind bacterial toxins, but senses host protein changes indirectly, similar to the plant guard model. These results indicate that pyrin exists in phosphorylated and inactive forms in the resting state, and is dephosphorylated and activated in response to RhoA inactivation caused by infection (Figure 1, right).

### 2.2. NLRP3 Inflammasome

The NLRP3 inflammasome secretes IL-1β/IL-18 in response to various stimuli, making it more likely to recognize host changes like pyrin rather than directly binding to the target. While many studies have reported the activation mechanism of NLRP3, little is known about how cells keep NLRP3 inactive in the resting state or how the NLRP3 inflammasome is negatively regulated (Figure 1, right). Here, we focused on the negative regulation of NLRP3 in the resting state.

## *3.* Negative Regulation of NLRP3

### 3.1. Molecular Inhibitors

#### 3.1.1. Heat Shock Protein 70 (HSP70)

HSP70 inhibits NLRP3 via direct interactions. The downregulation of HSP70 triggers NLRP3 inflammasome hyperactivation, while overexpression inhibits IL-1β secretion mediated by NLRP3. HSP70 likely maintains NLRP3 in its inactive form in the resting state [6].

#### 3.1.2. HSP90 and Suppressor of G2 Allele of SKP1 (SGT1)

HSP90 is a highly conserved and abundant chaperone that forms complexes with the co-chaperone SGT1. The HSP90-SGT1 complex functions as a co-chaperone for plant R proteins and maintains them in an inactive state. Given that NLRP3 functions as an indirect sensor, similar to plant R proteins in mammalian cells, it is plausible that the HSP90-SGT1 complex also plays a role in suppressing NLRP3 activation during the resting state. SGT1 has been implicated in the activation of nucleotide binding oligomerization domain-containing protein 1 (NOD1) and NLRP3 in mammals. Mayor et al. demonstrated that the HSP90-SGT1 complex plays a dual role in NLRP3 inflammasome activation. In its resting state, this complex binds to NLRP3 and maintains its inactive form. However, during the activation process, the inhibition of HSP90 suppressed NLRP3 inflammasome activation. Specifically, the knockdown of human SGT1 by siRNA or the chemical inhibition of HSP90 by geldanamycin abrogated NLRP3 inflammasome activity in THP-1 cells [7]. Similarly, NLRP3 inflammasome activation is abrogated in geldanamycin-treated human retinal pigment epithelial (RPE) cells [8]. These findings suggest that the HSP90-SGT1 complex not only suppresses NLRP3 inflammasome signaling in the resting state but also facilitates NLRP3 activation upon introducing activators. The involvement of the HSP90-SGT1 complex in NLRP3 inflammasome activity may indicate a link between the NLRP3 inflammasome and the plant guard model. In the subsequent sections of this paper, we will discuss the NLRP3 cage model, which has been shown to inhibit NLRP3 in the resting state but is also essential during the activation phase. This suggests a potential association between the cage model and the HSP90-SGT complex.

#### 3.1.3. Caspase Recruitment Domain-Containing Protein 8 (CARD8) [9]

CARD was initially identified as a protein–protein interaction motif involved in regulating apoptosis. It is also known for its function as a scaffolding molecule that induces inflammation by activating nuclear factor (NF) κB [10,11,12]. CARD8, also known as TUCAN, CARDINAL, or NDDP1, interacts with NLRP3 and significantly reduces IL-1β secretion [9]. In the resting state, NLRP3 is associated with CARD8 but not with Apoptosis-associated speck-like protein containing a CARD (ASC). However, upon introducing an activation signal such as ATP, NLRP3 associates with ASC instead of CARD8. Interestingly, CARD8 did not interact with cryopyrin-associated autoinflammatory syndromes (CAPS)-associated NLRP3 mutants, indicating that CARD8 negatively regulates NLRP3 by directly binding to it. Since CARD8 cannot bind to these mutants and suppress mutant NLRP3, the mutant NLRP3 easily forms an inflammasome complex and produces IL-1β, leading to autoinflammatory diseases. In addition, a study on the association between NLRP3 and Crohn’s disease reported that a loss-of-function CARD8 missense mutation promotes NLRP3 inflammasome activation. Serum levels of IL-1β in CARD8 mutant carriers were increased compared with healthy controls, and their monocytes secreted more IL-1β after activator treatment. This mutant did not bind to NLRP3, resulting in hyperactivation of NLRP3 [13]. These findings highlight the critical role of CARD8 in regulating NLRP3 inflammasome activation and its implications for autoinflammation and Crohn’s disease (Figure 2).

### 3.2. Structural Insights into NLRP3 Inactivation and Activation Mechanisms

#### 3.2.1. Closed Form 

Compan et al. reported that NLRP3 molecules are in spatial proximity to each other, with the N-terminus of one NLRP3 being closer to the C-terminus of an adjacent NLRP3 in the resting state [14]. They measured the conformational changes in NLRP3 using a Bioluminescence Resonance Energy Transfer (BRET)-based method, which enabled them to distinguish between the open and closed forms of NLRP3. Their findings demonstrated that potassium efflux converts the closed form of NLRP3 into its open form. Additionally, imiquimod, a potassium-independent NLRP3 activator, induces conformational changes in the open form. Therefore, NLRP3 activators induce conformational changes from the closed to the active form of NLRP3, a process that is not solely mediated by potassium efflux. However, the precise conditions promoting these conformational changes remain unclear.

#### 3.2.2. Cage Model

Wu et al. found that NLRP3 is maintained in an inactive form within a decamer cage-like structure in humans or in a 12-mer to 16-mer double cage form in mice [15]. Initially, NLRP3 shields the pyrin domain to prevent activation and forms a decameric cage structure, which requires the interaction of the LRR acidic loop with the adjacent LRR base sequence. Mutation of the acidic loop leads to the hyperactivation of NLRP3, suggesting that this structure is formed prior to NLRP3 activation [16]. Notably, this inactive NLRP3 structure was mainly localized in membrane compartments, consistent with previous reports, although the types of membranes to which NLRP3 binds vary, including the endoplasmic reticulum (ER), Golgi bodies, and mitochondria [17,18,19,20]. Without stimulation, NLRP3 cages are present on membranes that serve as platforms to recruit NLRP3 cages. Mutations in NLRP3 that cause a failure to form the cage structure inhibit inflammasome formation. Since this cage structure is also required for the activation of NLRP3, it simultaneously prevents abnormal activation in the resting state and promotes activation after activator treatment. Recently, Wu et al. also reported the cryo-EM structures of the NLRP3–NIMA Related Kinase 7 (NEK7)–ASC complex [21]. The fish-specific NACHT associated (FISNA) domain of NLRP3 is a critical mediator of both NLRP3 conformational changes and oligomerization, which is consistent with the defective NLRP3 activation observed with single mutations or complete or partial deletions of the FISNA domain. NLRP3 conformational change requires both ATP binding and the FISNA domain, likely because either domain alone is insufficient. Finally, they suggested that after the priming step undertaken by Toll like receptor (TLR) ligands such as lipopolysaccharide (LPS), NLRP3 expression is upregulated, and then an NLRP3 cage is formed on the membranes, particularly the trans-Golgi network (TGN), before stimulation. Moreover, because the HSP90-SGT1 chaperone complex, as discussed earlier, binds to NLRP3 LRRs and maintains them in an inactive form, it is possible that the HSP90-SGT1 complex is required for maintaining the double-ring cage structure of NLRP3.

### 3.3. The Membrane Association of NLRP3 Inflammasome 

As discussed in the previous section, various membranes have been proposed as platforms for NLRP3 activation. Recently, two studies further elucidated this concept by demonstrating that NLRP3 senses endosomal stress. These studies highlight changes in the composition of endosomes as a marker of NLRP3 activation and suggest that NLRP3 recognizes endosomal stress. One study reported that changes in endosomal composition indicate NLRP3 activation, suggesting that NLRP3 senses endosomal stress [22]. Similarly, another study emphasized the importance of the disruption of endocytic trafficking for NLRP3 activation [23], suggesting that NLRP3 is partially associated with membranes in the resting state. Upon receiving activation signals, NLRP3 translocates to membranes containing phosphatidylinositol 4-phosphate (PI4P), where it is activated. Alternatively, the composition of the membrane to which NLRP3 binds changes to include more PI4P. Chae et al. reported that phosphatidylinositol 4,5-bisphosphate was hydrolyzed into inositol trisphosphate and diacylglycerol upon receiving NLRP3 activation signals [24]. Therefore, the composition of phosphatidylinositol phosphates (PIPs) in the membrane may play a crucial role in maintaining NLRP3 in its resting state and regulating its activation. These findings suggest that dynamic changes in membrane PIP composition are not only closely associated with NLRP3 activation, but are also crucial for maintaining its inactivation (Figure 2).

### 3.4. Post-Translational Modifications (PTMs)

#### 3.4.1. SUMOylation

Barry et al. reported that NLRP3 is associated with the E3 ligase mitochondrial-anchored protein ligase (MAPL), and that it is SUMOylated (small ubiquitin-like modifier (SUMO)) in its resting state. Upon activation, the binding of MAPL to NLRP3 is inhibited, resulting in the deSUMOylation of NLRP3. The depletion of MAPL enhances inflammasome formation. Moreover, a mutation in NLRP3 (K689R), which impairs SUMOylation, leads to increased IL-1β production. Furthermore, the downregulation of the SUMO-specific proteases SUMO specific peptidase (SENP) 6 and SENP7 by siRNA suppressed NLRP3 inflammasome activation. These data indicate that MAPL SUMOylates NLRP3 at multiple sites, restraining inflammasome activation in the resting state [25]. In addition, SENP3 also deSUMOylates NLRP3 and blocks inflammasome assembly [26], suggesting that SUMOylation may loosen the closed or cage form of NLRP3, allowing it to transition into an activation-ready form. 

In the priming state, an increase in tripartite motif-containing protein (TRIM) 28 levels leads to its binding to NLRP3, where it promotes the SUMOylation of NLRP3. This SUMOylation process subsequently inhibits the ubiquitination and proteasomal degradation of NLRP3, thereby stabilizing NLRP3 and facilitating its activation [27].

#### 3.4.2. Ubiquitination

One of the most common methods used to inhibit NLRP3 activation is to induce its degradation by ubiquitination, ensuring that only low levels of NLRP3 exist in resting cells. In the resting state, NLRP3 is usually degraded by the ubiquitin-proteasome pathway, maintaining low expression levels, such that the basal level of NLRP3 does not form the inflammasome complex. Several E3 ligases have been reported to ubiquitinate NLRP3, including Membrane-Associated RING-CH-type Finger 7 (MARCH7) [28], Ariadne homolog 2 (ARIH2) [29], TRIM20 (also known as pyrin) [30], TRIM31 [31], casitas B-lineage lymphoma proto-oncogene b (CBLB) [32], F-box/LRR-repeat protein (FBXL2) [33], ubiquitin specific peptidase 5 (USP5) [32,34], and RING finger protein 125 (RNF125) [32]. These ligases function as negative regulators of NLRP3 [35]. Other studies have reported that NLRP3 is ubiquitinated in LPS-primed macrophages, and that its ubiquitination status significantly decreases after exposure to NLRP3 activators. Treatment with deubiquitinating enzyme (DUB) inhibitors increases NLRP3 ubiquitination and blocks its activation, indicating that ubiquitination plays a key role in suppressing NLRP3 activation [35]. Besides the degradation pathway, polyubiquitination affects NLRP3 activity. ARIH2 interacts with NLRP3 and induces poly-ubiquitination, reducing NLRP3 activity without causing degradation [29]. A recent study reported that cullin1 (CUL1) inhibits NLRP3 complex assembly by promoting K63-linked ubiquitination at the K689 site, potentially disrupting inflammasome formation by competing with the adaptor protein ASC without inducing degradation. Additionally, Ren et al. demonstrated that thiolutin, a zinc chelator that inhibits BRCC3 (a subunit of BRISC), suppresses the deubiquitination and activity of NLRP3 [36,37]. Xu et al. reported that the membrane-bound E3 ubiquitin ligase, glycoprotein (gp) 78, induces the mixed ubiquitination of NLRP3 and suppresses NLRP3 inflammasome activation. Macrophages from gp78 knockout mice secrete elevated IL-1β compared with wild-type mice [38]. Tang et al. have suggested that TRIM65 promotes the ubiquitination of NLRP3 through direct interactions. TRIM65 binds to the NACHT domain of NLRP3 and prevents NEK7 from binding to NLRP3 [39]. The inhibition of the deubiquitinases USP7 and USP47 attenuates NLRP3 inflammasome activation in macrophages [40,41]. Deubiquitinase YOD1 interacts with NLRP3, leading to the inhibition of NLRP3 inflammasome activation by removing a specific ubiquitin chain [42]. These reports collectively suggest that the ubiquitination of NLRP3 is essential for repressing its activation in the resting state. 

#### 3.4.3. cAMP

cAMP binds to regulatory subunits of enzymes and triggers conformational changes. Lee et al. found that cAMP directly binds to NLRP3 and suppresses inflammasome assembly. The downregulation of cAMP by KH7, which blocks cAMP synthesis, relieves this inhibition. The binding of cAMP to the constitutively active form of NLRP3, which causes CAPS, was much lower than that of wild-type NLRP3 [24]. Yan et al. found that dopamine (DA) inhibits NLRP3 through the dopamine D1 receptor (DRD1). DRD1 stimulates adenylate cyclase activity and cAMP production, thereby suppressing the NLRP3 inflammasome assembly [28]. Therefore, it is possible that the binding of cAMP to NLRP3 stabilizes the double-ring cage structure and prevents inflammasome complex assembly in the resting state.

#### 3.4.4. Phosphorylation

Spleen tyrosine kinase (SYK) [43,44,45], Death-associated protein kinase (DAPK) [46], Transforming growth factor β-activated kinase 1 (TAK1) [47], and Extracellular signal-regulated kinase 1 (ERK1) [48] are known to phosphorylate NLRP3. However, whether each kinase phosphorylates specific residues of NLRP3 remains unknown. The phosphorylation of NLRP3 at serine 5 inhibits its interaction with ASC, thereby blocking ASC speck formation. Protein Phosphatase 2 (PP2A) dephosphorylation activates the NLRP3 inflammasome. Scharl et al. reported that tyrosine 861, which is regulated by Protein Tyrosine Phosphatase Non-receptor type 22 (PTPN22), is crucial for NLRP3 activation. The knockdown of PTPN22 significantly decreases IL-1β secretion in response to NLRP3 agonists. Bone marrow-derived macrophages (BMDMs) from PTPN22 knockout mice produce less IL-1β compared with wild-type mice. Mechanistically, PTPN22 dephosphorylates tyrosine 861 of NLRP3, leading to its activation [49,50]. Thus, NLRP3 is likely to be phosphorylated at multiple sites in the resting state to prevent inflammasome assembly. Similarly, non-phosphorylated NLRP3 binds to ASC, resulting in inflammasome formation. The role of phosphorylation in regulating NLRP3 activity is complex and controversial. The phosphorylation of serine 295 by Protein Kinase D (PKD) activates NLRP3 [51], whereas phosphorylation by Protein Kinase A (PKA) inhibits NLRP3 activity by blocking its ATPase activity [52]. The PKD-mediated phosphorylation of NLRP3 in the Golgi induces its release from mitochondria-associated membranes, resulting in inflammasome formation. Conversely, PKA activation by Prostaglandin E2 (PGE2) phosphorylates serine 295, thereby suppressing NLRP3 inflammasome activation. Mutations in serine 295 that prevent phosphorylation are associated with CAPS, indicating that serine 295 phosphorylation inhibits NLRP3 activity in the resting state. Interestingly, tyrosine 861 is in the same region as serine 806 within the inactive NLRP3 cage structure, and is involved in LRR–LRR interactions. When NLRP3 is dephosphorylated at these sites, it promotes the activation of NLRP3 [50], suggesting that phosphorylation is crucial for maintaining the stability of the NLRP3 double-ring structure and preventing inflammasome complex formation in the resting state (Figure 3).

#### 3.4.5. Nucleotide Exchange

NLRP3 is a member of the STAND (signal transduction ATPases with numerous domains) family, which functions as a regulatory connection, integrating various signals. This family of proteins contains a conserved NOD (nucleotide-binding oligomerization domain) with ATPase activity. Generally, STAND family proteins exist in an inactive monomeric, ADP-bound state and, upon stimulation, are converted into ATP-bound, active, and multimeric forms [53].

The Nod module of Apoptotic Protease Activating Factor 1 (APAF1), a member of the STAND family, is in a closed conformation with the ADP molecule, which is deeply buried and non-exchangeable. The stability of this closed form depends on the interdomain interactions [54]. Although it is unclear whether the stability of the closed form of APAF1 is a common feature across all STAND family members, recent studies on the NLRP3 structure have suggested that the cage form of NLRP3 can be stabilized by internal LRR interactions, implying that NLRP3 may have similar characteristics. 

In the NLRP family, NLRP1, NLRP7, NLRP10, and NLRP12 have been reported to bind ATP, which is usually involved in their activation [55,56,57,58,59,60]. Therefore, it is plausible that NLRP3 exists in an ADP-bound inactive form and is converted to an active multimeric form by nucleotide exchange from ADP to ATP upon stimulation. Walker A and B motifs are well-known conserved sequences in the STAND protein family responsible for ATPase activity. Mutations in the Walker B motif may negatively affect ATP hydrolysis, resulting in a constitutively active NLRP3. This suggests that ATP binding stabilizes the active form of NLRP3 and is involved in turning off the NLRP3 activity after activation. Recent studies have reported that the sulfonylurea MCC950, a well-known NLRP3 inhibitor, binds to both the active and inactive states of NLRP3 through the Walker B motif, thereby blocking ATP hydrolysis and the assembly of NLRP3 [61]. MCC950 can also close the active open conformation of NLRP3, thereby preventing the conformational changes necessary for inflammasome assembly [62].

Recent studies have indicated that wild-type NLRP3 exhibits different hydrolytic activities depending on its conformation. In its resting state, NLRP3 binds to ADP in its inactive form. Reversible PTMs, such as phosphorylation and ubiquitination, convert inactive NLRP3 to a primed state that is not autoinhibited. This primed NLRP3 can then bind ATP and transition to an active state capable of forming an inflammasome complex [63]. Therefore, nucleotide exchange in NLRP3 is an early step in the activation of the NLRP3 inflammasome.

#### 3.4.6. Palmitoylation

Recent studies have highlighted the crucial role of palmitoylation in the regulation of the NLRP3 inflammasome. Specifically, the palmitoylation of NLRP3 at cysteine residue 844 (C844) by zinc finger DHHC-type palmitoyl transferase 12 (ZDHHC12) has been identified as a critical mechanism for preventing inappropriate inflammasome activation [63]. Additionally, NLRP3 activation is critically regulated through the palmitoylation of cysteine 126 (Cys126) by the enzyme ZDHHC7. This post-translational modification is pivotal for the proper functioning of the NLRP3 inflammasome, particularly in chronic inflammatory conditions [64].

### 3.5. Spatial Separation

NLRP3 primarily localizes to ER structures in the resting state, whereas the adaptor protein ASC exists in the cytosol. Upon activation, NLRP3 and ASC translocate to the perinuclear space and colocalize with the ER and mitochondria [20]. Other studies have suggested that, under resting conditions, ASC is observed in the mitochondria, cytosol, and nucleus, whereas NLRP3 is mainly confined to the ER [18]. It is likely that, in the resting state, NLRP3 and ASC are separated into different regions and migrate to the same region upon activation.

Several studies have reported that NEK7 binds to NLRP3 in its resting state [64,65,66]. However, these interaction data may be false positives owing to the disruption of cell compartments during cell lysis. In the resting state, NEK7 is localized in the centrosome, whereas NLRP3 is present in the cytosol [67]. This spatial separation of NLRP3 and NEK7 could be one of the mechanisms that block NLRP3 activation in the resting state.

### 3.6. Inhibitors Associated with Resting State (Table 1)

#### 3.6.1. MCC950

MCC950, also known as CRID3, is a diarylsulfonylurea-containing compound and one of the most potent and selective inhibitors of NLRP3. The first report on MCC950 was published in 2001 [68], and later, O’Neill’s group demonstrated the precise inhibitory function of MCC950 on the NLRP3 inflammasome in an in vivo mouse model [69]. In 2019, two back-to-back studies elucidated the molecular mechanism of MCC950 on the NLRP3 protein. Coll, R. C. et al. found that MCC950 directly binds to the Walker B motif in the NACHT domain and blocks ATP hydrolysis, thereby suppressing the ATPase activity of NLRP3 and preventing ASC oligomerization [61]. Tapia-Abellan, A. et al. suggested that MCC950 forces the NLRP3 “open” structure into a “closed” conformation that cannot form an inflammasome complex. In the BRET assay, while both BRET signals from MCC950-treated samples and from the resting state indicated a closed structure of NLRP3, the structure of NLRP3 in the resting state was easily converted to the open, active form upon stimulation. In contrast, the NLRP3 structure in the MCC950-treated samples did not change to the open form in response to activators, suggesting that the closed structure induced by MCC950 treatment differed from that in the resting state [62].

Sharif et al. reported that MCC950 and ADP stabilized NLRP3 and NLRP3/NEK7 complexes in a thermal shift assay [70]. Based on these results, Wu et al. purified the NLRP3 protein with or without MCC950, and obtained similar NLRP3 structures in both cases. Furthermore, since the NLRP3 protein must be maintained in an inactive form during preparation, MCC950 was also used for this purpose. According to Hochheiser et al., MCC950 interacts with residues from five different subdomains (HD1, HD2, WHD, NBD, and trLRR), leading to a conformational rearrangement of the NACHT domain in response to stimulators, stabilizing their conformation, and suppressing NLRP3 inflammasome activation. This suggests that MCC950 stabilizes the ADP-bound inactive form of NLRP3 [71].

#### 3.6.2. CY-09 (Glitazone) [72]

Jiang et al. identified a new effective inhibitor of NLRP3 that demonstrated dramatic effects in both in vivo and ex vivo experiments. CY-09 binds to the cysteine 172 residue of the Walker A motif in the NACHT domain of NLRP3, thus blocking its oligomerization and activation. NLRP3 contains both Walker A and B motifs, with the Walker A motif essential for ATP binding and the Walker B motif responsible for ATP hydrolysis. Both motifs are necessary for the ATPase activity of NLRP3. Since the binding of CY-09 to NLRP3 is affected by Walker A mutation but not by Walker B mutation, and CY-09 suppresses ATP binding to NLRP3 in a dose-dependent manner, CY-09 inhibits NLRP3 activity by disrupting its ATPase activity through direct interaction; therefore, it will maintain the inactive cage form of NLRP3.

#### 3.6.3. Tranilast

Tranilast (N-[3′,4′-dimethoxycinnamoyl]-anthranilic acid) is a tryptophan metabolite analog that exhibits suppressive effects in mouse models of the disease [73,74]. Moreover, tranilast selectively inhibited the NLRP3 inflammasome without affecting the AIM2 and NLRC4 inflammasomes. Tranilast interacts with the NACHT domain of NLRP3, thereby hindering its oligomerization similarly to CY-09. However, tranilast treatment did not affect the interaction between NLRP3 and NEK7 or ASC, nor does it impact the ATPase activity of NLRP3. Notably, tranilast blocked NLRP3-NLRP3 interactions, which are important for the cage-like structure of NLRP3. This suggests that, while the cage structure prevents the abnormal activation of NLRP3 in the resting state and promotes activation upon stimulation, tranilast can suppress the activation step. Further studies are required to determine whether tranilast treatment affects the cage-like form of NLRP3 [75].

#### 3.6.4. OLT1177

OLT1177, also known as dapansutrile, is a β-sulfonyl nitrile compound that specifically inhibits the NLRP3 inflammasome without affecting the AIM2 or NLRC4 inflammasomes. Marchetti et al. reported that OLT1177 treatment reduced CXCL1, MPO (myeloperoxidase), and IL-6 levels in a mouse model of LPS-induced systemic inflammation [76]. The Fluorescence Resonance Energy Transfer assay results indicate that OLT1177 treatment significantly reduced ASC oligomerization. In an in vitro ATPase activity assay, OLT1177 inhibited the ATPase activity of the recombinant NLRP3 protein. Since the NACHT domain of NLRP3 exhibits ATPase activity and OLT1177 suppresses this activity, the authors concluded that OLT1177 inhibits the ATPase activity of NLRP3 through a direct interaction with the NACHT domain. However, to obtain more convincing results, further in vivo confirmation is required. It is likely that OLT1177 stabilizes the closed inactive form of NLRP3 through direct interactions. Notably, unlike other NLRP3 inhibitors, OLT1177 is safe and exhibits no organ toxicity at any of the doses tested. ijms-25-09018-t001_Table 1Table 1Inhibitors associated with resting state.NameCompoundTarget SiteMechanism of ActionAssociation with Resting StateRef.MCC950 (CRID3)Diarylsulfonylurea-containing compoundWalker B motif in NACHT domainReducing the ATPase activity of NLRP3 Forces the NLRP3 “open” structure into a “closed” conformation [56]HD1, HD2, WHD, NBD, and trLRR[66]CY-09Glitazone derivateCysteine 172 of Walker A motif in NACHT domainDisrupting the ATPase activity of NLRP3 Possibility of maintaining a closed, inactive form[67]TranilastTryptophan metabolite analogNACHT domainBlocking NLRP3-NLRP3 interactionPossibility of blocking cage formation[70]OLT1177β-sulfonyl nitrile compoundNACHT domainBlocking ATPase activityPossibility of stabilizing a closed, inactive form[71]OridoninBioactive constituent of the medical plant *Rabdosia rubescens*Cysteine 279 in NACHT domainBlocking NLRP3-NEK7 interactionPossibility of affecting cage structure and migration[77]MNSSYK inhibitor groupNACHT and LRR domainReducing the ATPase activity of NLRP3 Possibility of stabilizing a cage structure[78]IFN58Acylate and Acrylamide DerivativesCys319 in NACHT domainReducing the ATPase activity of NLRP3 Possibility of affecting cage structure[79]IFN39Acylate and Acrylamide DerivativesUnknownReducing the ATPase activity of NLRP3 Affecting conformational change and possibility of influencing closed form[80]


#### 3.6.5. Oridonin

Oridonin is a major bioactive constituent of the medicinal plant *Rabdosia rubescens*, which is used extensively in traditional medicine. It has been studied for its positive effects on various diseases, including Alzheimer’s disease, Crohn’s disease, osteoarthritis, and cancer [77,78,79,80,81]. Mechanistic studies have shown that oridonin blocks ASC oligomerization but does not affect ROS production or potassium efflux, two upstream signaling events, suggesting that oridonin directly affects NLRP3 activation. Oridonin blocks the interaction between NLRP3 and NEK7, which is essential for NLRP3 inflammasome assembly. In contrast to OLT1177 and Tranilast, Oridonin did not inhibit NLRP3-NLRP3 interaction or ATPase. Instead, Oridonin bound to the cysteine 279 residue of NLRP3 via covalent bond formation. Unlike CY-09, Oridonin irreversibly binds to NLRP3. These data indicate that oridonin directly binds to NLRP3 and disrupts NEK7 binding [82].

Further research is needed to elucidate how oridonin forms a direct covalent bond with NLRP3 using recent structural data on NLRP3, whether this covalent bond formation affects the inactive cage-like structure in the resting state, and how it influences the process of NLRP3 migration to NEK7 for binding through Histone Deacetylase (HDAC) and microtubules [67].

#### 3.6.6. MNS (3,4-Methylenedioxy-β-nitrostyrene) [83]

While screening a kinase inhibitor library to identify the kinases involved in NLRP3 activation, He et al. discovered that MNS, a member of the SYK inhibitor group, suppressed NLRP3 activity. However, they found that SYK was not essential for NLRP3 activation. Additionally, MNS did not affect potassium efflux, a key upstream signaling factor for NLRP3 inflammasome activation, nor did it affect the activation of other inflammasomes such as AIM2 and NLRC4. Mechanistically, MNS specifically blocked ASC speck formation by directly interacting with the NACHT and LRR domains of NLRP3. MNS also reduces the ATPase activity of NLRP3 in a dose-dependent manner. Structurally, the nitrovinyl side chain of the MNS is essential for its inhibitory activity. Since MNS has the potential to inhibit NLRP3 activity by covalently binding to a cysteine residue in the ATP-binding site of the NACHT domain, it is highly likely that MNS stabilizes the inactive cage-like form of NLRP3 and prevents conformational changes or translocation even in the presence of an activation signal. Further studies on the relationship between NLRP3 in the resting state and the MNS may provide more convincing data regarding its inhibitory function.

#### 3.6.7. Acrylate and Acrylamide Derivatives

Bertinaria et al. developed a series of electrophilic compounds to prevent NLRP3-dependent pyroptosis. They demonstrated that molecules acting as Michael acceptors efficiently prevented NLRP3 activation. However, owing to their high reactivity, these molecules are cytotoxic. Therefore, safer compounds, namely, acrylamide derivatives, have been developed. These derivatives inhibited NLRP3 ATPase activity. IFN58 was found to successfully suppress NLRP3 activity. In silico predictions suggest that IFN58 binds to cysteine 319 of NLRP3, which is located in the ATPase catalytic pocket [84]. However, further confirmation is required. They also reported that IFN39, another candidate, is a non-toxic, irreversible NLRP3 inhibitor [85]. In a BRET assay, IFN39 reduced the steady-state BRET signal of NLRP3, indicating that IFN39 negatively affected its conformation in the resting state. After stimulation, IFN39 also affected conformational changes at a specific stage independent of potassium efflux. Interestingly, because IFN39 affects NLRP3 in both the resting and activated states, it may influence the inactive cage-like structure. Understanding how IFN39 affects the conformation of NLRP3 and its inactive cage-like structure in the resting-state is pivotal for the development of new drugs (Figure 4).

## 4. Discussion

NLRP3 is a well-known and extensively studied inflammasome, and numerous studies have reported its activation mechanism. These studies focused on identifying new factors that activate the NLRP3 inflammasome and essential binding partners during activation, such as ASC and NEK7. However, little is known about how NLRP3 remains inactive in the resting state and how it is negatively regulated without stimulation after priming.

Similar to plant guard proteins, pyrin is activated by recognizing changes or modifications caused by bacteria without directly interacting with PAMPs or DAMPs. Because NLRP3 does not directly detect DAMPs or PAMPs, it is plausible that NLRP3 is regulated in a manner similar to pyrin, following a mammalian guard model. Furthermore, a review of pyrin and NLRP3 inflammasomes recognizing intracellular homeostatic changes [86], and a research article suggesting that NLRP3 and pyrin are similarly activated through HDAC and microtubules [83] reinforces this prediction.

The fact that NLRP3 is inhibited in a manner similar to the plant guard hypothesis and pyrin in its resting state is significant for the following reasons. 

First, recent studies have reported that NLRP3 is associated with membranes in an inhibited state, and is subsequently activated in response to specific membrane environments [22,23]. However, if NLRP3 directly binds to membranes, this alone does not sufficiently explain why it would be activated in some membranes while being inhibited in others. This suggests the potential involvement of additional inhibitory proteins that regulate NLRP3’s membrane association. These proteins may play a crucial role in modulating NLRP3’s activation or inhibition depending on the membrane context. For example, pyrin’s activity is inhibited by its interaction with 14-3-3 proteins, indicating that similar inhibitory proteins may interact with NLRP3 in its resting state. Therefore, it is essential to identify these proteins using techniques such as TurboID and mass spectrometry to further explore the proteins associated with NLRP3 in its resting state.

Second, within this framework, it is also plausible to consider the existence of inhibitory proteins that stabilize the cage model of NLRP3. These proteins could be key determinants of NLRP3’s activation and inhibition by binding to and dissociating from NLRP3. Consequently, future experiments should focus on distinguishing the membranes associated with NLRP3 during its inhibited state from those during its activated state. Identifying the novel proteins that bind to NLRP3 in these different membrane environments could provide critical insights into the mechanisms governing NLRP3 activation.

Given that the abnormal activation of NLRP3 has been linked to various diseases, including neutrophilic asthma, Alzheimer’s disease, inflammatory bowel disease, and atherosclerosis, biopharmaceutical companies are now developing NLRP3 inhibitors as potential treatments for these conditions. To date, most NLRP3 inhibitors developed so far are known to inhibit NLRP3 activity by binding to the NACHT domain, maintaining NLRP3 in its resting state, even upon stimulation. In addition, the PTMs of NLRP3 in the resting state have been reported in recent studies. Because inhibiting PTMs can affect many other cellular responses, it is crucial to investigate the mechanisms that specifically block NLRP3 PTMs in order to develop new drugs. Therefore, understanding the negative regulation of NLRP3 in the resting state is crucial for the identification of therapeutic targets and the development of novel strategies to treat diseases associated with NLRP3 dysregulation.

## Figures and Tables

**Figure 1 ijms-25-09018-f001:**
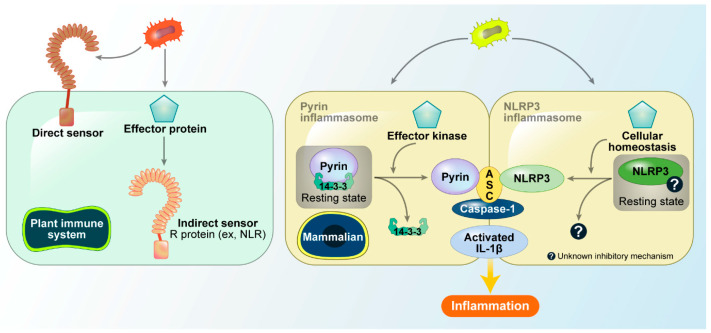
Guard hypothesis in plant and mammalian cells. There are two types of plant immune responses: PTI (Pattern-Triggered Immunity) and ETI (Effector-Triggered Immunity). In PTI, direct sensors can recognize pathogen-associated molecular patterns. ETI, on the other hand, is an evolutionary response to pathogen strategies, where an indirect sensor detects changes in the intracellular environment caused by pathogen effectors. This allows plants to detect a wide range of effector proteins despite lacking an adaptive immune system. Mammalian cells may also have indirect sensors like ETI. Pyrin and the NLRP3 inflammasome are examples of such sensors, capable of detecting intracellular changes. For Pyrin, the 14-3-3 protein binds to it, preventing conformational changes that would lead to activation. In the case of the NLRP3 inflammasome, an unknown inhibitory protein may function similarly to 14-3-3, maintaining NLRP3 in an inactive state.

**Figure 2 ijms-25-09018-f002:**
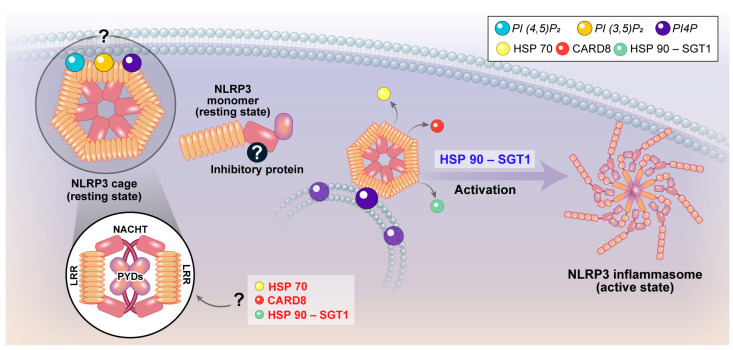
Negative regulation of NLRP3 in resting state. After the priming step, NLRP3 adopts a cage-like structure, predominantly associated with membranes. However, the specific membrane that facilitates this cage formation remains unidentified. During this phase, HSP70, HSP90-SGT1, and CARD8 may play crucial roles in preventing the full activation of the NLRP3 inflammasome. Upon receiving an activation signal, NLRP3 undergoes a conformational change, resulting in inflammasome assembly. Notably, HSP90-SGT1 serves a dual role, both inhibiting NLRP3 activation and functioning as a crucial platform during the activation phase. The question mark indicates an unknown mechanism that has yet to be elucidated.

**Figure 3 ijms-25-09018-f003:**
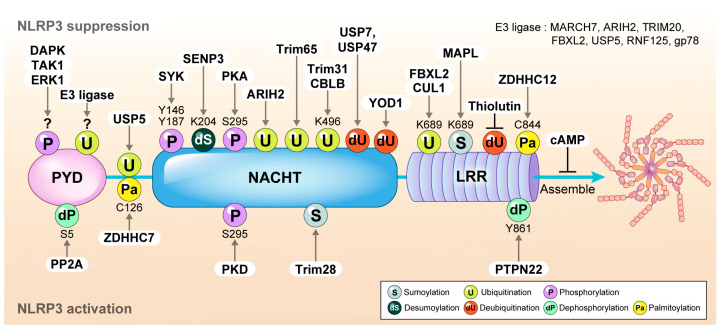
Post-translational modifications in resting state. The domain organization of NLRP3 and its post-translational modifications play critical roles in the regulation of the NLRP3 inflammasome. Upward regulatory modifications lead to the suppression of NLRP3 activity, while downward regulatory modifications promote the activation of NLRP3 inflammasome.

**Figure 4 ijms-25-09018-f004:**
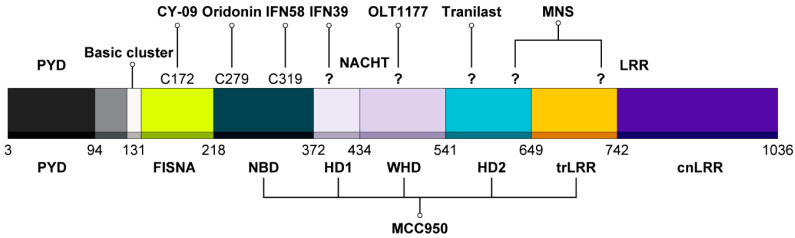
Target site of inhibitors associated with the resting state. NLRP3 consists of eight subdomains: the PYD (3–94) and a linker segment (95–130); the FISNA (131–218), NBD (219–372), HD1 (373–434), WHD (435–541), and HD2 (542–649) subdomains forming the NACHT domain; and a transition LRR (650–742) and canonical LRR (743–1036) within the LRR domain. Each subdomain corresponds to specific target sites for NLRP3 inhibitors. The question mark indicates an unknown mechanism that has yet to be elucidated.

## Data Availability

Data available in a publicly accessible repository.

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
