# Peer review of "NLRP3 Negative Regulation Mechanisms in the Resting State and Its Implications for Therapeutic Development"

_ijms, 2024, doi:10.3390/ijms25169018_

Round 1
Reviewer 1 Report
Comments and Suggestions for Authors
The NLRP3 (NOD-, LRR-, and pyrin domain-containing protein 3) is a component of the innate immune system that plays a critical role in the formation and activation of the NLRP3 inflammasome, which is responsible for the activation of inflammatory responses, particularly the production of pro-inflammatory cytokines such as IL-1β and IL-18. In their submitted manuscript, YeJi Kim et al. highlights the molecular mechanisms that regulate NLRP3 in its resting state and discusses how 20 targeting this inhibitory mechanism can lead to novel therapeutic strategies for NLRP3-related dis-21 eases, which is necessary. To me, this work allows us to gain a better understanding of how NLRP3 is activated and inhibited. I only have the following minor questions.
1. Please modify figure 2 to include more details of interactions of HSP70, 90, CARD8.
2. Figure 2, please also shows NLRP3 monomer as resting state.
3. Some modifications/corrections in the manuscript may be needed.
Comments on the Quality of English LanguageN/A
Author Response
The NLRP3 (NOD-, LRR-, and pyrin domain-containing protein 3) is a component of the innate immune system that plays a critical role in the formation and activation of the NLRP3 inflammasome, which is responsible for the activation of inflammatory responses, particularly the production of pro-inflammatory cytokines such as IL-1β and IL-18. In their submitted manuscript, YeJi Kim et al. highlights the molecular mechanisms that regulate NLRP3 in its resting state and discusses how 20 targeting this inhibitory mechanism can lead to novel therapeutic strategies for NLRP3-related dis-21 eases, which is necessary. To me, this work allows us to gain a better understanding of how NLRP3 is activated and inhibited. I only have the following minor questions.
- Please modify figure 2 to include more details of interactions of HSP70, 90, CARD8.
As requested, I have redrawn Figure 2 and included additional explanations in the legend. Thank you for your precise feedback.
- Figure 2, please also shows NLRP3 monomer as resting state.
While redrawing Figure 2, we labeled the monomer inside the membrane as the resting state. Thank you.
- Some modifications/corrections in the manuscript may be needed.
I have added the necessary revisions throughout the document, with the changes marked in red text.
Thank you.
Reviewer 2 Report
Comments and Suggestions for Authors
The review entitled: NLRP3 Negative Regulation Mechanisms in the Resting State and its Implications for Therapeutic Development “provides a comprehensive and insightful analysis of the mechanisms underlying the negative regulation of the NLRP3 inflammasome. The title effectively captures the study's main purpose, and the abstract concisely summarizes the key objectives, findings, and potential implications. The article is undoubtedly publishable, given its good summary of the topic, although several areas could be improved.
Let’s start with the introduction to the article, which is well-written and offers a detailed review of the role of the inflammasome in the immune system, with a focus on NLRP3. The authors clearly articulate the gap in the current literature regarding the regulation of NLRP3 in the resting state, highlighting their study's relevance and importance. The article itself examines the mechanisms of negative regulation, discussing various molecular inhibitors and regulatory proteins, such as HSP70, HSP90, and CARD8, which play critical roles in maintaining NLRP3 in an inactive state. At the same time, the discussion of therapeutic implications is particularly compelling, as it discusses the regulation of NLRP3 and presents opportunities for potential inhibitors such as MCC950 and CY-09. This connection highlights the translational impact of their findings, which provides a strong argument for further research in this area.
While on the subject of discussion, it would be good if the discussion of post-translational modifications such as phosphorylation and SUMOylation could be expanded to include more details on the specific sites and conditions that affect these modifications and their precise impact on NLRP3 activity.
The figures and diagrams used throughout the article effectively illustrate the key concepts, although, in my opinion, additional explanations should accompany these figures, as most figures simply describe what they represent. The reader would benefit from a more extensive description below the figure.
A final suggestion to the authors is to expand the section on future research directions with an indication of specific studies that could be used to verify the hypotheses presented in the paper.
In summary, the paper itself is well organized, with a clear and logical progression, allowing readers to easily follow the complex discussion. Technical terms are clearly defined, ensuring accessibility for readers with varying levels of knowledge in molecular biology. However, expansion of some issues, as well as the suggestions above, would make it easier for readers to read.
Author Response
The review entitled: NLRP3 Negative Regulation Mechanisms in the Resting State and its Implications for Therapeutic Development “provides a comprehensive and insightful analysis of the mechanisms underlying the negative regulation of the NLRP3 inflammasome. The title effectively captures the study's main purpose, and the abstract concisely summarizes the key objectives, findings, and potential implications. The article is undoubtedly publishable, given its good summary of the topic, although several areas could be improved.
Let’s start with the introduction to the article, which is well-written and offers a detailed review of the role of the inflammasome in the immune system, with a focus on NLRP3. The authors clearly articulate the gap in the current literature regarding the regulation of NLRP3 in the resting state, highlighting their study's relevance and importance. The article itself examines the mechanisms of negative regulation, discussing various molecular inhibitors and regulatory proteins, such as HSP70, HSP90, and CARD8, which play critical roles in maintaining NLRP3 in an inactive state. At the same time, the discussion of therapeutic implications is particularly compelling, as it discusses the regulation of NLRP3 and presents opportunities for potential inhibitors such as MCC950 and CY-09. This connection highlights the translational impact of their findings, which provides a strong argument for further research in this area.
While on the subject of discussion, it would be good if the discussion of post-translational modifications such as phosphorylation and SUMOylation could be expanded to include more details on the specific sites and conditions that affect these modifications and their precise impact on NLRP3 activity.
-> As per your suggestions, I have added the known phosphoryation, SUMOylation, and ubiquitination sites to Figure 3. For regions where the specific sites are not yet identified, I did not mark them. Thank you.
The figures and diagrams used throughout the article effectively illustrate the key concepts, although, in my opinion, additional explanations should accompany these figures, as most figures simply describe what they represent. The reader would benefit from a more extensive description below the figure.
--> To assist readers understanding, I have added legends to all figures. Thank you for your insightful feedback.
A final suggestion to the authors is to expand the section on future research directions with an indication of specific studies that could be used to verify the hypotheses presented in the paper.
--> I have added content related to future studies in the discussion section, with the new additions marked in red text. This addition has significantly strengthened the overall discussion. Thank you for your feedback.
In summary, the paper itself is well organized, with a clear and logical progression, allowing readers to easily follow the complex discussion. Technical terms are clearly defined, ensuring accessibility for readers with varying levels of knowledge in molecular biology. However, expansion of some issues, as well as the suggestions above, would make it easier for readers to read.